# Human embryonic stem cells secrete macrophage migration inhibitory factor: A novel finding

Yanzhao Wei[1,2,3], Xiaohan Zheng[1,3], Ting Huang[1,3], Yuanji Zhong[1,3], Shengtong Sun[1,3], Xufang Wei[1,3], Qibing Liu[4], Tan Wang[1,3]*, Zhenqiang Zhao[1,3]*

1 Department of Neurology, First Affiliated Hospital of Hainan Medical University, Hainan, China, 2 Department of Human Functioning, Department of Health Services, Logistics University of Chinese People's Armed Police Force, Tianjin, China, 3 Key Laboratory of Brain Science Research & Transformation in Tropical Environment of Hainan Province, Hainan Medical University, Hainan, China, 4 Department of Pharmacy, Hainan Medical University, Hainan, China

☯ These authors contributed equally to this work.
* zhenqiang.zhao@qq.com (ZZ); wanghy19980204@foxmail.com (TW)

**Data Availability Statement:** All relevant data are within the manuscript and its Supporting Information files.

## Abstract

Macrophage migration inhibitory factor (MIF) is expressed in a variety of cells and participates in important biological mechanisms. However, few studies have reported whether MIF is expressed in human Embryonic stem cells (ESCs) and its effect on human ESCs. Two human ESCs cell lines, H1 and H9 were used. The expression of MIF and its receptors CD74, CD44, CXCR2, CXCR4 and CXCR7 were detected by an immunofluorescence assay, RT-qPCR and western blotting, respectively. The autocrine level of MIF was measured via enzyme-linked immunosorbent assay. The interaction between MIF and its main receptor was investigated by co-immunoprecipitation and confocal immunofluorescence microscopy. Finally, the effect of MIF on the proliferation and survival of human ESCs was preliminarily explored by incubating cells with exogenous MIF, MIF competitive ligand CXCL12 and MIF classic inhibitor ISO-1. We reported that MIF was highly expressed in H1 and H9 human ESCs. MIF was positively expressed in the cytoplasm, cell membrane and culture medium. Several surprising results emerge. The autosecreted concentration of MIF was 22 ng/mL, which was significantly higher than 2 ng/mL-6 ng/mL in normal human serum, and this was independent of cell culture time and cell number. Human ESCs mainly expressed the MIF receptors CXCR2 and CXCR7 rather than the classical receptor CD74. The protein receptor that interacts with MIF on human embryonic stem cells is CXCR7, and no evidence of interaction with CXCR2 was found. We found no evidence that MIF supports the proliferation and survival of human embryonic stem cells. In conclusion, we first found that MIF was highly expressed in human ESCs and at the same time highly expressed in associated receptors, suggesting that MIF mainly acts in an autocrine form in human ESCs.

**Funding:** This work was supported by a grant from National Natural Science Foundation of China (grant numbers 81860238); Hainan Provincial Natural Science Foundation of China (grant number 821RC694); Major Research and Development Project of Hainan Province of China (grant number ZDYF2018233); Scientific research projects in Colleges and Universities of Hainan Province of China (grant number HYYS2021A19); Project of Hainan Province Clinical Medical Center. The funders had no role in study design, data collection and analysis, decision to publish, or preparation of the manuscript.

**Competing interests:** The authors have declared that no competing interests exist.

## Background

Embryonic stem cells (ESCs) originate from the inner cell mass of mammalian blastocysts [1]. Human ESCs were first reported by researchers in 1998 [2]. The pluripotent state of ESCs allows them to differentiate into any other somatic cell type except the placenta [3]. These important characteristics of ESCs have great application potential in helping understand cell development, drug development and disease treatment [4]. At present, human ESCs have been used for cell repair and treatment of various diseases, such as spinal cord injury, age-related macular degeneration and type I diabetes [5]. Effective maintenance of ESC stemness is a critical factor in achieving these key applications. Some intracellular signaling pathways such as Wnt/β-catenin [6], BMP/Smad [7], LIF/ GP130 /STAT3 [8], FGF/ERK [9] and PI3K/Akt [10] are involved in maintaining the self-renewal activity of ESCs. In addition to the genes in the above pathways, some transcription factors such as Nanog, Oct4 and Sox2 also play very important roles in maintaining the self-renewal activity and multidirectional differentiation characteristics of ESCs [11]. Studies have found that ESCs have a strong immunosuppression effect, which can effectively weaken the innate immune response of inflammatory cytokines [12], thus avoiding cellular immune damage. In addition, the physiological hypoxia microenvironment is also a key factor for ESCs to maintain self-renewal activity and pluripotency [13]. These key features of self-renewal and differentiation properties of ESCs are strictly controlled by different mechanisms such as multiple signaling pathways, translation factors and epigenetic changes [14]. This complex network is extremely important for differentiation and development. Although remarkable progress has been made in stem cell biology in recent years, our understanding of the complexity of ESCs is limited [15]. Therefore, it is necessary to explore the potential molecules and signaling pathways that affect the multidirectional differentiation, proliferation and survival of ESCs.

Macrophage migration inhibitory factor (MIF) was originally a cytokine isolated from the supernatant of cultured T lymphocytes, which can seriously interfere with macrophage migration [16]. Studies have found that macrophages are not the only cellular source, many other cell types such as endothelial cells, eosinophils and neuronal cells also express this important cytokine [17].

MIF exerts its biological functions in an autocrine and paracrine manner via the receptors CD74, CD44, CXCR2, CXCR4, and CXCR7 [18–22], which may trigger the activation of multiple signaling pathways including Akt and ERK, and ultimately promotes the activation of pro-inflammatory cytokines and transcription factors required for cell cycle regulators to promote cell migration, proliferation [23–27]. In addition, MIF can regulate autophagy [28, 29] and interfere with the survival of senescent cells. MIF is also a downstream target of hypoxia-inducible factor-1 α (HIF-1α), which interferes in many biological activities by regulating innate immunity [30]. Previous studies have shown that MIF is expressed in a variety of stem cells and is involved in regulating cell proliferation and differentiation including cancer stem cells, neural stem cells, cardiac stem cells, cartilage endplate stem cells and adipose-derived stem cells [31–33].

In this study, we cultured human ESCs to explore whether human ESCs expressed MIF and its important protein receptors CD74, CD44, CXCR2, CXCR4 and CXCR7, investigate which receptors bind to MIF, and preliminarily explore the effect of MIF on the proliferation and survival of human ESCs, hoping that the results obtained could enrich and better our understanding about the differentiation and development mechanisms of human ESCs and provide basic theoretical support for the effective application of human ESCs.

## Materials and methods

### Cultivation of human ESCs

Matrigel (BD Bioscience, USA) working solution was coated in the culture plate. H1 and H9 human ESC lines (Chinese Academy of Sciences, Shanghai China) were inoculated in mTeSR™1 (STEM CELL, Canada) medium and cultured in a cell incubator with a volume fraction of 5% $CO_2$ at 37˚C. The cell culture medium was routinely changed. When the cell colonies became larger, the centers became dense and bright (compared to their edges) and the adjacent colonies began to fuse, digestion and passage were carried out. The cells used in the experiment were in the logarithmic growth phase.

### ELISA experiment

The Matrigel working solution was coated in cell culture wells of 12-well plates. $5 \times 10^4$ H1 and H9 cells in the logarithmic growth phase were inoculated in mTeSR medium and cultured in a cell incubator with a volume fraction of 5% $CO_2$ at 37˚C. The number of cells was calculated at 24, 48 and 72 h. The supernatant was taken. The expression of MIF in the supernatant was determined by using the ELISA detection kit (JL11770, Jianglai Biology, China).

### RNA isolation and qRT-PCR

RNA was isolated from H1 and H9 human ESCs using TRIzol reagent (15596026, Invitrogen, USA). Reverse transcriptase (11123ES60, Yi Sheng Bio, China) was used for reverse transcription. qPCR was performed by using Hieff® qPCR SYBR Green Master Mix (Low Rox Plus) kit (11202ES08, Yi Sheng Bio, China). ACTIN was used as an internal reference gene for qRT-PCR and the primers are shown in Table 1. Data were analyzed using ΔΔCT method.

### Immunofluorescence experiments

The 24-well cell culture plates were divided into H1 and H9 groups. Cell climbing sheets were placed and coated with Matrigel working solution. The cells were rinsed twice with PBS, incubated with 4% paraformaldehyde on ice for 15 min, rinsed three times with pre-cooled PBS, added with 1 mL blocking solution (P0260, Beyotime, China) and incubated for 30 min, incubated with antibody MIF (1:800, GTX53741, GeneTex, USA), CD74 (1:500, ab108393, abcam, UK, CD44 (1:400, Ab243894, abcam, UK), CXCR2 (1:100, 20634-1-AP, Wuhan Sanying, China), CXCR4 (1:500, ab181020, abcam, U.K), CXCR7 (1:800, GTX100027, GeneTex, USA) overnight at 4˚C, rinsed three times with PBS, then incubated with a secondary antibody goat anti-mouse IgG-H & L (Alexa Fluor® 488) (1:800, ab150113, Abcam, U.K) or goat anti-rabbit IgG-H & L (Alexa Fluor® 594) (1:800, ab150080, Abcam, UK) at room temperature for 2 h, rinsed three times with PBS. DAPI staining solution was added to cover the cells and incubated for 5 min. After sealing and drying, the cells were placed on the Nikon fluorescence microscope for observation.

**Table 1. The primer sequence for qPCR sequences.**

| name | Primer sequences (fwd/rev) |
| --- | --- |
| H-MIF-S | GCAGAACCGCTCCTACAGCAAG |
| H-MIF-A | TGGCCGCGTTCATGTCGTAATAG |
| H-ACTIN-S | CACCCAGCACAATGAAGATCAAGAT |
| H-ACTIN-A | CCAGTTTTTAAATCCTGAGTCAAGC |

## Western-blot experiments

H1 and H9 cells were cultured normally and total protein was extracted with RIPA high-efficiency lysate. The protein concentration was measured according to the instructions of BCA Protein Quantification Kit (G2026-200T, Servicebio, China). A pipetting gun was used to accurately add proteins of the same mass into the compression gel and the voltage and current conditions were determined according to the molecular weight of different proteins. The protein was separated at 120 V voltage for some time and transferred to the PVDF membrane. The membrane was kept at 250mA constant for 90 min, sealed with sealing solution (P0023B-100mL, Beyotime, China), and incubated with antibody MIF (1:1000, sc-53915, Santa Cruz, USA), CD74 (1:1000, ab108393, Abcam, UK), CD44 (1:1000, Ab243894, Abcam, UK), CXCR2 (1:1000, 20634-1-AP, Proteintech, China), CXCR4 (1:1000, Ab181020, Abcam, UK), CXCR7 (1:800, GTX100027, GeneTex, USA) at 4°C overnight, rinsed with PBS for three times, and then incubated with horseradish peroxidase (HRP) -conjugated secondary antibody goat anti-rabbit IgG (H+L) (1:1000, A0208, Beyotime, China) or goat anti-mouse IgG (H+L) (1:1000, A0216, Beyotime, China) at room temperature for 2 h. A hypersensitive ECL chemiluminescence kit (P0018S, Beyotime, China) was used to detect protein bands, and the relative protein content of each group was analyzed by ImageJ software.

## Flow cytometry analysis

In this experiment, the expression of related factors on the cell surface was detected by the indirect method of primary antibody coupled with fluorescence secondary antibody, and the detection system was used at 500 μL. The number of H9 cells was about $1.5 \times 10^6$, and the cell precipitation was obtained by digestion and centrifugation. The cell precipitate was suspended with 1 mL 10% FBS solution on ice, and the suspension was respectively added into 1.5 mL EP tubes labeled CXCR2 and CXCR7. The corresponding primary antibody was added. The cells were incubated for 30 min at room temperature away from light, washed with pre-cooled PBS and centrifuged twice, added with fluorescent secondary antibody diluted with 3% BSA/PBS solution, incubated at 4°C for 25 min under dark conditions, washed with pre-cooled PBS, and centrifuged twice. The cell precipitate was suspended with 3%BSA/PBS solution 500 μL each, transferred to 4°C and stored away from light for inspection.

## Immunoprecipitation (Co-IP) assay

H1 and H9 were cultured in T25-culture bottle, and washed with cold PBS, followed by lysing using IP lysis buffer. After centrifuging at 12,000×g for 10 min at 4°C, 100 μL of the supernatant was added with 20 μL protein A/G beads (SC-2003, Santa Cruz, USA) and incubated with the 2ug antibody overnight at 4°C. The next day, the immunoprecipitants were washed three times and boiled with 2×SDS Loading Buffer. Do western blotting assay according to the protocol above.

## Immunofluorescence performed by confocal microscopy analysis

According to the above-mentioned immunofluorescence experimental method, the analysis was carried out under the immunofluorescence confocal microscope.

## Cell proliferation and toxicity assay

The experiment was divided into control group, MIF (NBp2-35005, NOVUS, USA) group, CXCL12 (Ab259416, Abcam, U.K) group, and ISO-1 (HY-16692, MCE, USA) group. MIF group was divided into 30 ng/mL group, 100 ng/mL group, and 300 ng/mL group. CXCL12

group was divided into 10 ng/mL group, 40 ng/mL group, and 160 ng/mL group. ISO-1 group was divided into 12 µM group, 24 µM group, 48 µM group, 96 µM group and 192 µM group. Each group was set up with three repeated wells, 96-well cell culture plates were prepared, and related grouping labels were made. After 24-, 48- and 72-h cultures, 10 µL CCK-8 reagent (BS350B, BioSharp, China) was added, and absorbance was measured at 450nm with the microplate reader. The experiment was repeated three times.

### Statistical analysis

All experimental data were statistically analyzed by GraphPad Prism8.0.2 analysis software. All experimental data are expressed as X ± S. t-test was used for comparison between two groups, and one-way analysis of variance (One-way ANOVA) was used for comparison between multiple groups. All data were from at least 3 independent experiments. $P<0.05$ indicated that the difference between groups was statistically significant.

## Results

### MIF expression in H1 and H9 human ESCs

We conducted a preliminary exploratory study to detect the expression of the MIF gene in H1 and H9 lines by qRT-PCR, and then verify the result by immunochemical fluorescence (ICF) and Western blot. Finally, the autocrine amount of MIF in H1 and H9 lines was detected by ELISA. It was found that MIF mRNA was expressed in H1 and H9 lines, showing no significant difference between the two cell lines (Fig 1A). As shown in Fig 1B for ICF detection, MIF exhibited strong green fluorescence on H1 and H9 cells and was distributed densely in the cell membrane and sparsely in the cytoplasm. Western blot results showed that MIF was expressed in both H1 and H9 cells. Compared with tubulin, the expression of MIF protein was slightly lower, and there was no significant difference between H1 and H9 cells (**Fig 1C and 1D**).

MIF secretion in H1 and H9 cells was detected by ELISA. With the culture time increasing, the cell culture medium became clearer under the light microscope, a small number of floating dead cells were seen, and the number of adherent cells with normal morphology was increased as compared with that before, and no obvious differentiation was observed. The number of H1 and H9 cells increased exponentially at 24, 48 and 72 h, and the number of cells in different lines was similar at the same time, but there was no significant difference. ELISA results showed that the content of MIF in the culture medium was relatively constant independent of the culture time and cell number, and there was no significant difference in secretion level between the two cell lines (**Fig 1E**).

### Expression of MIF-related receptors in human ESCs

To detect the expression of MIF related receptors in human ESCs, H1 and H9 human ESCs were cultured in this study for ICF and Western blot assay, respectively. ICF results showed that CD74, CD44, CXCR2, CXCR4 and CXCR7 were all expressed in H1 and H9 human ESCs, with the expression of CXCR2 and CXCR7 receptors predominating. It was found that the expression of CXCR2 was significantly stronger than that of CXCR7, showing no significant difference between H1 and H9 cell lines (**Fig 2A**). Western blot results showed that compared with the internal reference protein, CXCR2 had the highest expression level, followed by CXCR7, while the expression levels of CD74, CD44 and CXCR4 were significantly lower. There was no statistically significant difference in the expressions of these receptors between H1 and H9 cell lines (**Fig 2B and 2C**).

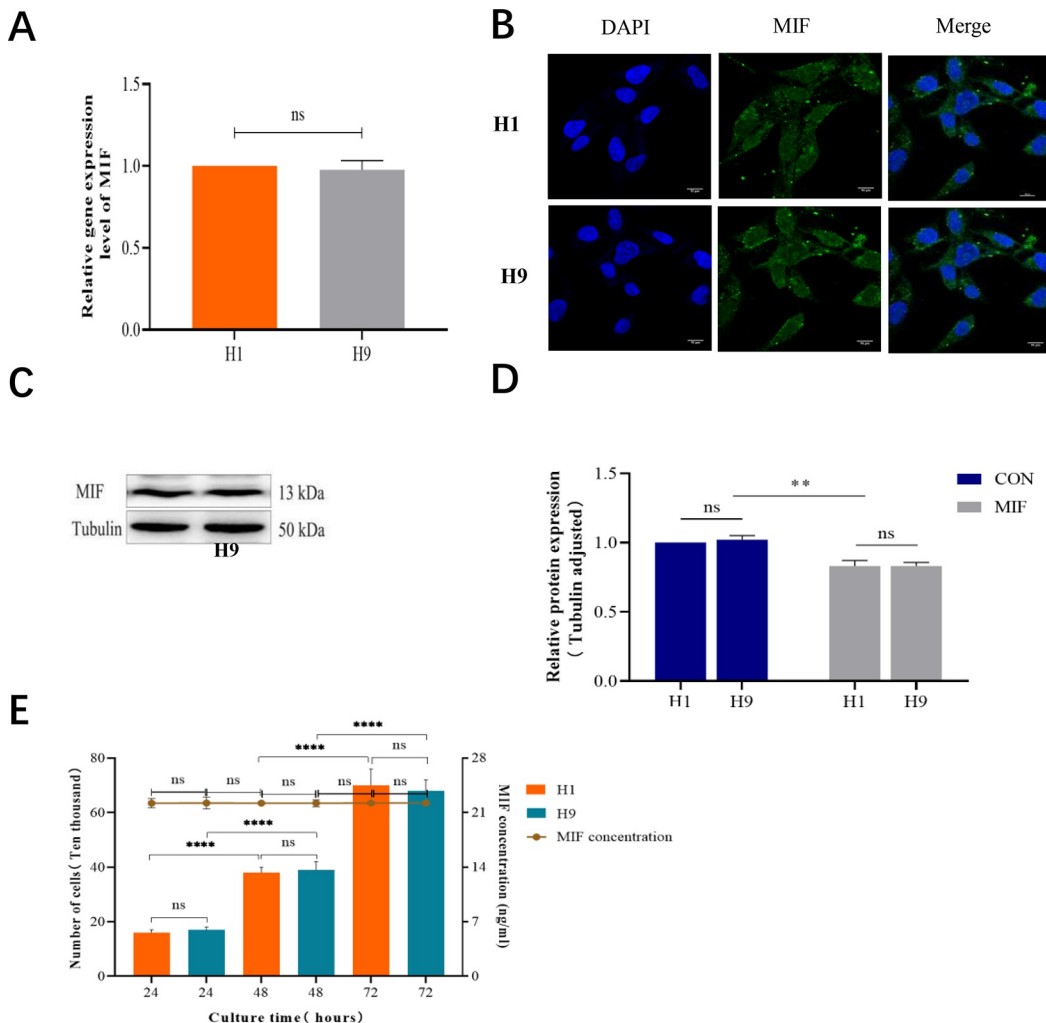

**Fig 1. Expression of MIF in H1 and H9 human embryonic stem cell (ESC) lines. A**. qRT-PCR analysis of MIF in H1 and H9 human ESCs, $P > 0.05$, n = 3; **B**. The expression of MIF in H1 and H9 human ESCs as detected by ICF; the nucleus is blue, and the depth of green fluorescence indicates the expression level of MIF in ESCs, the deeper the fluorescence color, the denser the distribution and the stronger the expression of the target protein, with a magnification of 1200 and a scale of 10 μm; **C**. The expression of MIF in H1 and H9 human ESCs as shown by Western blot; **D**. Semi-quantitative analysis of protein expression, compared with Tubulin, $**P < 0.01$, n = 3. E. MIF content in cell medium of H1 and H9 human ESCs at different times. Compared with H1, $P > 0.05$. Compared with 24 h, $***P < 0.001$, $****P < 0.0001$.

## Surface expression of the CXCR2 and CXCR7 receptors

To verify the interaction between MIF and receptors CXCR2 and CXCR7, the expression of receptors CXCR2 and CXCR7 on the cell membrane of human H9 embryonic stem cells was detected by FCM. The results showed that CXCR7 but not CXCR2 was expressed on the cell membrane of H9 cells (**Fig 3**).

## Interaction between MIF and receptor

To further determine the acting receptors of MIF, Co-IP experiments were conducted. Firstly, MIF antibody was used as bait protein and immunoprecipitation method was used to detect whether it could precipitate CXCR2 and CXCR7 proteins. The results showed that MIF and CXCR7 were mutually immunoprecipitated (**Fig 4A and 4C**). No evidence was found that

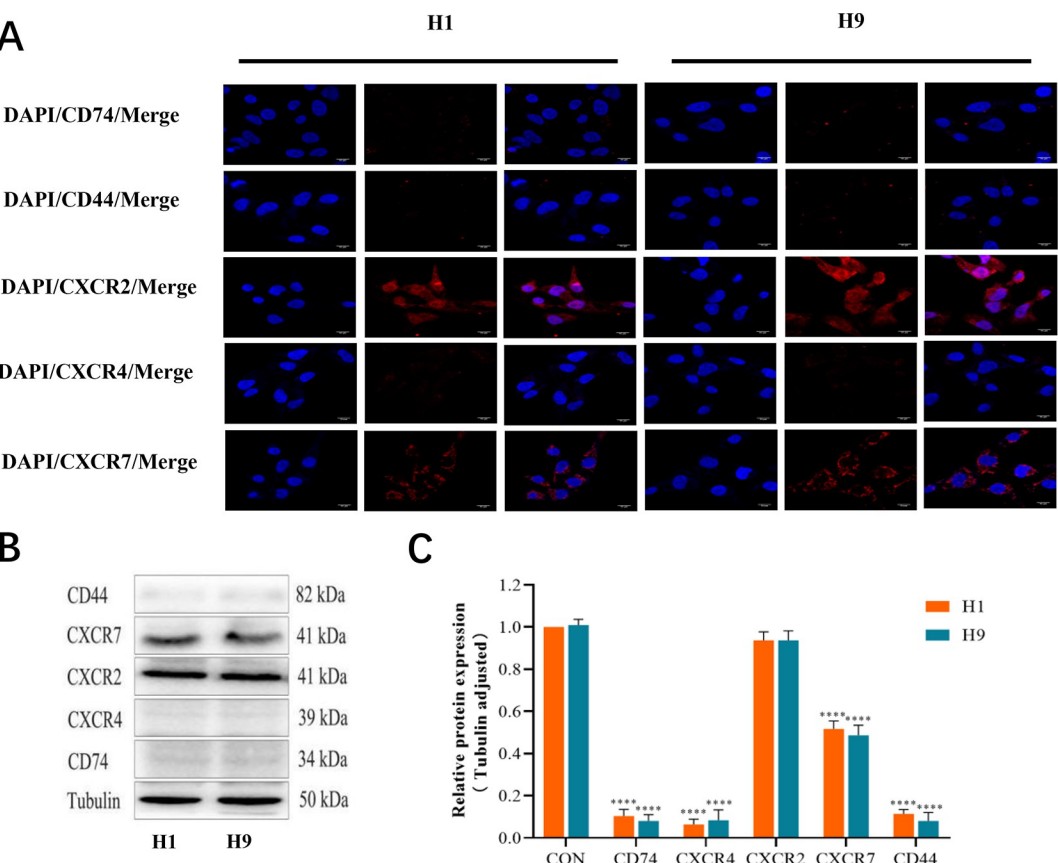

**Fig 2. Expression of MIF-related receptors in H1 and H9 cells. A**. The expression of MIF-related receptors in human H1 and H9 embryonic stem cells was detected by ICF; DAPI stained nuclei showed blue fluorescence, where the depth of red fluorescence indicates the MIF-related receptors CD74, CD4, CD44, CXCR2, CXCR4 and CXCR7, the deeper the red, the denser the distribution, indicating the higher the related receptor expression level, with a magnification of 1200 and a scale of 10m; **B**. Western blot was used to detect the expression of MIF-related receptors in human H1 and H9 embryonic stem cells; **C**. Semi-quantitative analysis of protein expression, compared with Tubulin, ****$P$ <0.0001, n = 3.

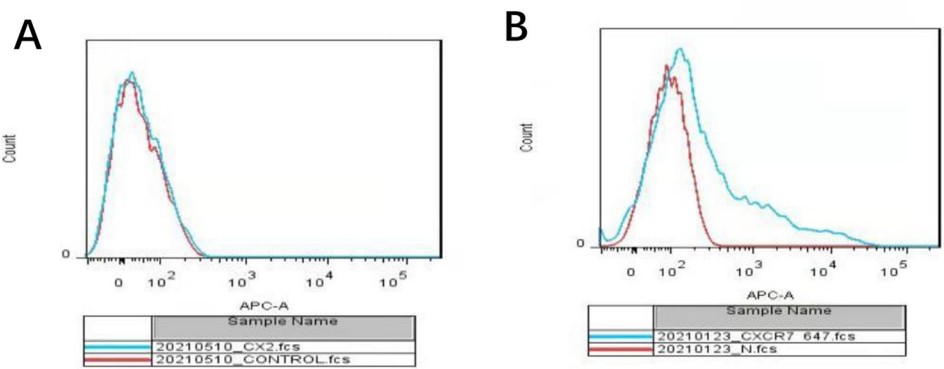

**Fig 3. Detection of the expression of CXCR2 and CXCR7 receptors on the membrane of H9 cells by FCM. A**. FCM showed no expression of CXCR2 on the membrane of H9 cells; **B**. FCM showed a positive expression of CXCR 7 on the cell membrane of H9 cells.

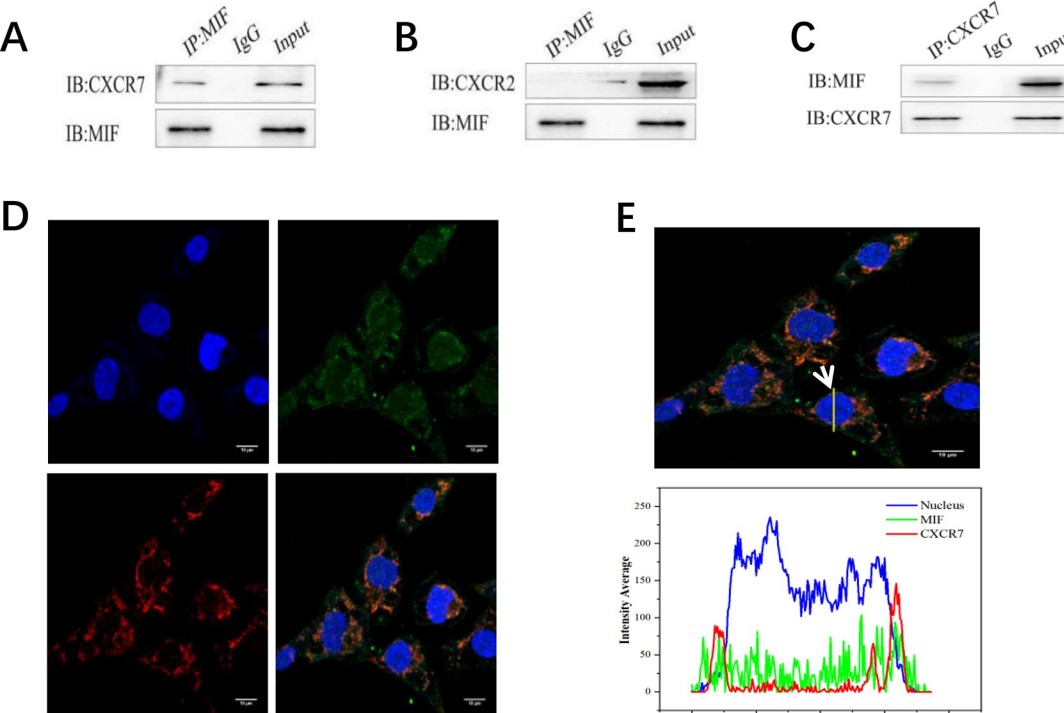

**Fig 4. Interaction of MIF with its receptors CXCR2 and CXCR7. A**. MIF antibody was used as bait protein, and the IB result showed that CXCR7 was successfully precipitated; **B**. MIF antibody was used as bait protein, and the IB result showed that CXCR2 could not be successfully precipitated; **C**. CXCR7 antibody was used as bait protein to reverse MIF precipitation, and the IB results showed that MIF was successfully precipitated. IB was the target protein as detected by Western blot, IP was immunoprecipitation (using relevant immunoprecipitation antibody), IgG was the negative control group, and Input was the positive control group. **D**. Nucleus (blue fluorescence), MIF (green fluorescence), CXCR7 (red fluorescence), magnification of 1200, scale of 10m; **E**. Nucleus (blue fluorescence), MIF (green fluorescence) and CXCR7 (red fluorescence) image composite, fluorescence quantitative analysis (white tip).

MIF precipitated CXCR2 (**Fig 4B**). Co-IP experiments indicated that MIF is mainly bound to the receptor CXCR7 in human embryonic stem cells. To support this conclusion, immunofluorescence stainings were performed using MIF and CXCR7 antibodies. Finally, Photoshop software was applied to overlay three fluorescence images with different colors. In the combined images, the expression positions of MIF and CXCR7 in the cell membrane were found to be duplicated, indicating that there existed spatial co-localization of the two proteins (**Fig 4D**). Fluorescence quantitative analysis of the combined images (arrows in the figure) showed that MIF was expressed in both the membrane and cytoplasm of H9 cells, while CXCR7 was mainly expressed in the membrane (**Fig 4E**).

### Effects of MIF, CXCL 12 and ISO-1 on the proliferation and survival of human ESCs

Based on the existing research conclusion, we mainly explored the role of MIF in human ESC stem cell proliferation and survival. Firstly, we used different concentrations of MIF to incubate cells and detected cell viability by CCK-8 assay. Compared with the control group, cell proliferation activity did not increase with the MIF concentration increasing, and there was no statistically significant difference between the groups with different MIF concentrations (**Fig 5A**). Then, we incubated cells with different concentrations of the MIF competitive ligand

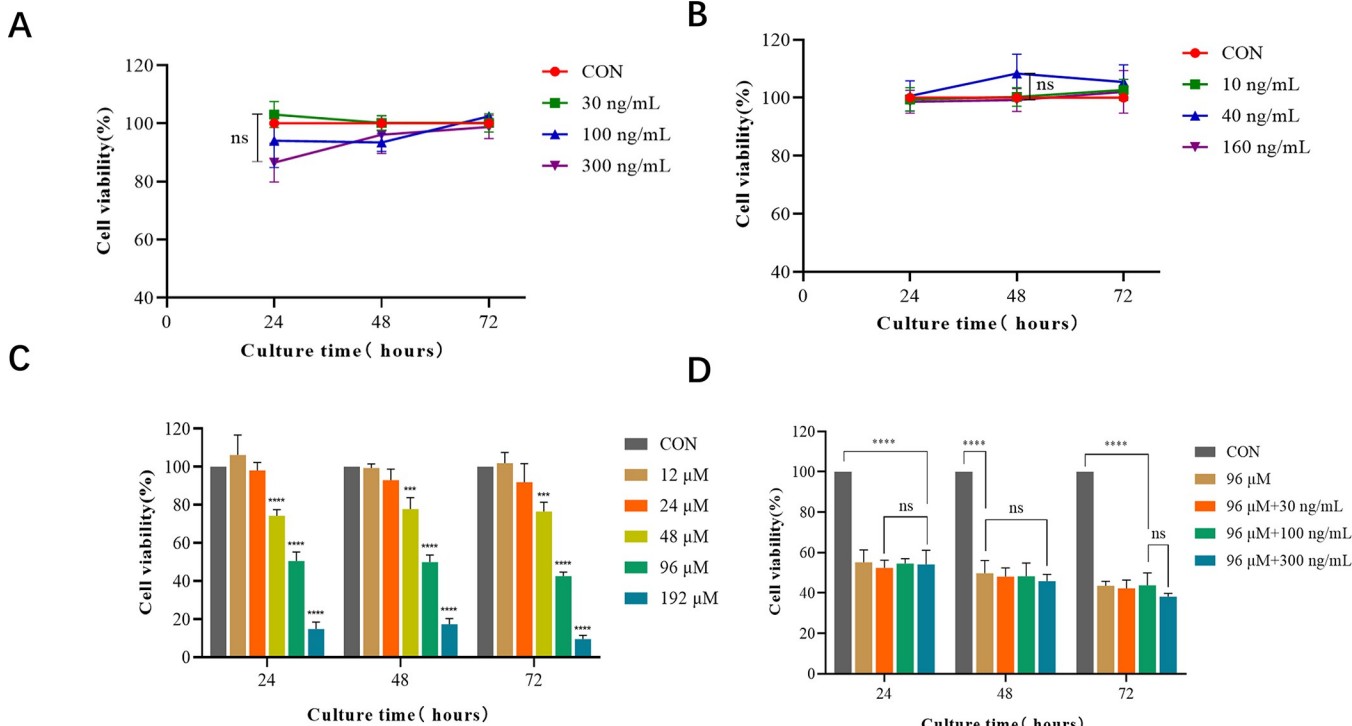

**Fig 5. Effects of MIF, CXCL12 and ISO-1 on human ESC proliferation and survival. A**. Cells were incubated with different concentrations of MIF for 24, 48 and 72 h, and the cell viability was detected by CCK-8 assay, showing no significant difference ($P>0.05$); **B**. Cells were incubated with different concentrations of CXCL12 for 24, 48 and 72 h, and cell viability was detected by CCK-8 assay, showing no significant difference ($P>0.05$); **C**. Cells were incubated with different concentrations of ISO-1 for 24, 48 and 72 h, and cell viability was detected by CCK-8 assay, compared with the control group, ***$P<0.001$, ****$P<0.0001$; **D**. Cells were co-incubated with 96 μM ISO-1 and different concentrations of MIF simultaneously for 24, 48 and 72 h, and cell viability was detected by CCK-8 assay, compared with the control group, ***$P<0.001$, ****$P<0.0001$.

CXCL12 and measured cell activity by CCK-8 assay. Compared with the control group, there was no significant difference in cell activity in each experimental group (**Fig 5B**). To further explore the effect of MIF on human ESC proliferation and survival, we incubated cells with different concentrations of the MIF classical inhibitor ISO-1 and detected cell activity by CCK-8 assay. Compared with the control group, the experimental groups of low concentrations of ISO-1(12 μM, 24 μM) had little effect on cell viability of H9 cells, while the medium and high concentrations (48 μM, 96 μM and 192 μM) significantly affected cell proliferation and survival in a dose-dependent manner (**Fig 5C**). To explore whether this biological effect of ISO-1 was caused by inhibiting MIF, we co-incubated the cells with ISO-1 and MIF simultaneously. The cell activity of 96 μM ISO-1 was around 50.58%, 49.93% and 42.50% at 24, 48 and 72 h, respectively. Therefore, it was used as an appropriate inhibitor dose in subsequent experiments. We co-incubated cells with 30 ng/mL, 100 ng/mL and 300 ng/mL MIF, and detected cell activity by CCK-8 assay. Compared with the control group, the cell viability was significantly reduced in each experimental group. However, there was no significant difference in cell viability between the experimental groups (**Fig 5D**).

## Discussion

Previous studies have shown that MIF is a pleiotropic cytokine produced by a variety of organs and cell types including neural stem cells, cardiac stem cells, cartilage endplate stem cells and adipose-derived stem cells. However, there is no study reporting the expression of ESCs and

secretion of important cytokines from SCs. There is also a lack of research on the biological functions of ESCs. To the best of our knowledge, this is the first study that investigated the expression, secretion and interaction of MIF and its receptors in human ESCs, as well as the effect of MIF on human ESC proliferation and survival.

MIF has been implicated in the pathogenesis of multiple organ-specific autoimmune diseases including type 1 diabetes, rheumatoid arthritis, multiple sclerosis, Guillain–Barr'esyndrome, Crohn's disease, autoimmune myocarditis, glomerulonephritis, hepatitis, thyroiditis, and psoriasis [34]. Although high expression of MIF is associated with the pathogenesis of several autoimmune diseases, MIF is also a highly conserved cytokine constitutively expressed in a variety of mammalian tissues under normal physiological conditions [35]. We first found that MIF was highly expressed in human ESCs in the cell membrane and cytoplasm. ELISA results showed that the secretion of human embryonic stem cells was about 22 ng/ml, which was much higher than the normal human serum concentration of 2–6 ng/ml [36]. In addition, it was independent of the culture time and cell number, indicating that there is a strict physiological regulation mechanism of MIF expression and secretion in human ESCs. Previous studies have shown that the expression and secretion of MIF are entirely dependent on cell types and stimulation [37]. In general, Gram-positive bacterial exotoxins, lipopolysaccharides, hypoxia conditions, stress or tissue damage can promote the secretion and release of MIF [38]. We know that the hypoxic microenvironment is necessary for stem cells to maintain self-renewal and pluripotency. HIF-1α is a key cellular regulator in response to hypoxia, which drives transcriptional regulation of HIF-1α and promotes enhanced expression of MIF [39]. This may be a major factor in the high expression of MIF in human ESCs. In addition, unlike MIF that is similar to the serum concentration observed in cerebrospinal fluid (CSF) due to the contact between nerve cells and cerebrospinal fluid [40], the closed local environment of ESCs may also be one of the main reasons for the significantly high expression of MIF. Importantly, our test results showed that there was no significant difference in the secretory expression of MIF between human H1 and H9 ESCs and they also expressed key receptors for MIF, which fully showed that MIF may play an important role in the differentiation of human ESCs by autocrine or paracrine participation in important life activities, and is an indispensable key cytokine for the ontogeny of early life.

D-dopachrome tautomerase (D-DT), a second member of the MIF superfamily, has been identified. D-DT shares significant homology with MIF, with 27% and 35% amino acid sequence identity in mice and humans, respectively. The 3D structure of D-DT is nearly identical to MIF with a barrel-shaped homotrimer. Similar to MIF, D-DT also conserves a vestigial enzymatic activity for converting D-dopachrome to DHICA in vertebrate organisms [28]. Like MIF, D-DT is produced by many tissue and immune cell types, and its circulating levels in blood are similar to MIF [41]. MIF and D-DT appear to exhibit unidirectional or cooperative activity in endotoxemia, some cancers, chronic obstructive pulmonary disease (COPD), and cardiac ischemia/reperfusion injury and heart failure [42]. From a pharmacological and clinical point of view, the nonredundant biological properties of MIF and D-DT anticipate potential synergisms from their simultaneous inhibition [43]. Therefore, we speculate that DDT may also be secreted in the stroma of human embryonic stem cells and has a coordinated effect with MIF.

MIF mainly binds to four receptors (CD74, CXCR2, CXCR4 and CXCR7). CD74 is the core receptor by which MIF exerts its action. CD74 knockout or application of anti-CD74 antibody can severely affect the MIF effect of cells expressing CD44, CXCR2, CXCR4 or CXCR7. However, we found that this important receptor as well as CD44 and CXCR4 receptors was not significantly expressed in human ESCs, but mainly expressed the receptors CXCR2 and CXCR7, and the expression of CXCR2 was mainly restricted to neutrophils and endothelial cells [44]. Previous studies have shown that CXCR2 required binding to CD74 to function

normally [25]. This is also supported by the finding that MIF did not undergo coimmunopre-cipitation with CXCR2 in human ESCs. Since no positive expression of CXCR2 on the cell membrane of human ESCs was found in this study, we believe that there may be a cytosolic internalization similar to the CD74 receptor [44], MIF may trigger the internalization of the CXCR2 receptor. CD74, the most well-studied receptor, is a type II transmembrane glycopro-tein similar to the histocompatibility complex (MHC) II and is mainly highly expressed on the cell surface of some immune cells such as B cells and macrophages [44]. Because CD74 has no typical signal transduction cytoplasmic domain, it cannot mediate MIF signal transduction as a separate receptor. However, its signal transduction can benefit the auxiliary receptor path-way, in which the recruitment and activation of non-receptor tyrosine kinases of CD44 play a key role. Studies have shown that CD74 and CD44 have significant interactions and can jointly form immunoprecipitation complexes [45–47], which proves the importance of CD44 auxil-iary receptor in the role of CD74. We found that CD74 and CD44 were poorly expressed in human ESCs, which may be different from previous understanding. CD74 may not play a key role in the role of MIF in human ESCs, which was also confirmed by the low expression of CD44. Both CD74 and CXCR4 binding or the triploids formed by CD74, CXCR4 and CXCR7 are the main shape of the MIF receptors [26, 48]. The low expression of CXCR4 in human ESCs supports the function of CXCR4 as a receptor complex, as CD74 remains a key partner for CXCR4 to function and a key partner in playing the role of CXCR4. However, the role of CD74 in mediating MIF signaling is not entirely indispensable. One report suggested that MIF activated the PI3K-Akt signaling pathway only through CXCR7 [49]. We also demonstrated that CXCR7 receptor was highly expressed in human ESCs, and CXCR7 receptor mainly existed in cell membrane. CXCR7 and MIF can form an immunoprecipitation complex, sug-gesting that MIF may have a similar action mechanism in human ESCs and platelets, and can interact with receptor CXCR7 independently without the classical coreceptor CD74. This fur-ther indicates the independence of CXCR7 receptor and its vital importance for MIF, though extensive research is required to verify our finding and conclusion.

One of the most important roles of MIF is the ability to promote cell proliferation and inhibit apoptosis [22, 47, 50]. Previous studies have shown that MIF can promote stem cell sur-vival and proliferation by increasing Akt, mitogen-activated protein kinase (MAPK) and AMPK phosphorylation [51–53]. We first incubated cells with different concentrations of exogenous MIF but did not find enhanced cell proliferation and survival after addition of exogenous MIF. This is primarily achieved with MIF by binding the receptor CD74, and the binding to MIF can also trigger CD74 internalization and deliver it to the endocytic compart-ment [47]. The CD74 intracellular domain (ICD) is released into the cytoplasm. The CD74-ICD can be transferred to the nucleus [59], where CD74-ICD actively regulates the activity of nuclear factor B (NF-B) and Runt-related transcription factor (RUNX) by binding with them, and affects cell proliferation and survival by upregulating the anti-apoptotic gene Bcl-xl [47]. The main reason for the absence of similar physiological effects in human ESCs may be the weak expression of CD74 because CD74 acts mainly as a pro-survival receptor [50]. However, it has been shown that the direct binding of MIF and CXCR7 can stimulate B cell migration, ERK1 / 2 signaling, and the activation of the zeta chain-associated protein kinase (ZAP7) [26]. But this requires the intervention of the receptors CD74 and CXCR4, and subsequent studies found that CXCR7 alone could bind MIF to induce platelet survival and cell proliferation through the PI3K / Akt signal transduction pathway [60]. We therefore pro-pose that MIF may bind to the CXCR7 receptor alone to exert biological effects. Previous stud-ies have demonstrated that CXCL12 is also a high-affinity ligand for CXCR7 [54, 55]. We incubated the cells using different concentrations of CXCL12 to see whether the ligand CXCL12 affected human ESC proliferation and survival by competing for the site of MIF

action. The results demonstrated that exogenous CXCL12 did not exert effects on human ESC proliferation and survival, suggesting that the main role of MIF in human ESCs may not lie in promoting their proliferation and survival. To verify this conjecture, we used different concentrations of MIF inhibitor (ISO-1), knowing that ISO-1 is often used as a standard inhibitor of MIF to inhibit the tautomerase activity of MIF and block the MIF-induced signaling cascade by preventing the binding of MIF to its surface receptor [56]. It was found that cellular activity gradually decreased with increasing exogenous ISO-1. We used an appropriate concentration of ISO-1 to incubate cells with different concentrations of MIF, and found that MIF did not reverse ISO-1, suggesting that MIF may not act in human ESCs, as was the case with the exogenous MIF that did not promote the survival of adipostem cells [57].

Studies have shown that MIF deficiency in the embryonic stage will seriously affect the normal development of the respiratory system, cause respiratory dysfunction, and reduce the fetal survival rate [58, 59]. It has been demonstrated that MIF plays an important role during embryonic development, especially in cells of the nervous system, and that can promote the proliferation and differentiation of neural progenitors [33]. In addition, MIF has a keto-enol tautomerase and thiol protein oxidoreductase activity [60, 61]. This may be related to the detoxification process of catecholamine degradation products, so MIF may be involved in the protection of human ESCs and lineage differentiation. As several specific MIF inhibitors are being developed and a semispecific inhibitor, ibidulast is already in the clinical setting [62]. The current discovery of the potential role of MIF produced by human embryonic stem cells and MIF in embryonic development makes it imperative to realize that MIF inhibition during pregnancy should be carefully evaluated.

Our study has some limitations. First, we only focused on the highly expressed receptors CXCR2 and CXCR7 without investigating other weakly expressed receptors, knowing that all of them may participate in mediating the key biological effects of MIF. In addition, the high level of CXCR2 is still a problem that can not be ignored, which may mediate the key biological effects of MIF.

## Conclusion

MIF is an important cytokine that has attracted increasing attention due to its involvement in multiple important biological functions. Studies on the cellular function of MIF participate in mediation of many complex biological processes. We found for the first time that MIF was highly expressed in human ESCs and at the same time highly expressed in associated receptors, suggesting that MIF mainly acts in an autocrine form in human ESCs. In addition, we found that exogenous MIF did not affect the proliferative survival of human ESCs, but what biological effects that MIF participates in remains an issue to be further investigated.

## Supporting information

**S1 Raw images.**
(PDF)

## Author Contributions

**Conceptualization:** Tan Wang, Zhenqiang Zhao.

**Data curation:** Yanzhao Wei, Xiaohan Zheng, Yuanji Zhong, Xufang Wei.

**Formal analysis:** Yanzhao Wei, Xiaohan Zheng, Ting Huang.

**Funding acquisition:** Zhenqiang Zhao.

**Investigation:** Yanzhao Wei, Xiaohan Zheng, Xufang Wei.

**Methodology:** Yanzhao Wei, Xiaohan Zheng, Yuanji Zhong, Qibing Liu.

**Project administration:** Yanzhao Wei, Xiaohan Zheng, Tan Wang, Zhenqiang Zhao.

**Resources:** Qibing Liu, Tan Wang, Zhenqiang Zhao.

**Software:** Yanzhao Wei, Xiaohan Zheng, Ting Huang, Yuanji Zhong.

**Supervision:** Yanzhao Wei, Xiaohan Zheng, Zhenqiang Zhao.

**Validation:** Ting Huang, Yuanji Zhong, Shengtong Sun.

**Visualization:** Ting Huang, Yuanji Zhong, Shengtong Sun, Xufang Wei.

**Writing – original draft:** Yanzhao Wei, Zhenqiang Zhao.

**Writing – review & editing:** Xiaohan Zheng, Tan Wang, Zhenqiang Zhao.

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
