## [Decision Letter · Decision Letter 0]

17 Apr 2023

PONE-D-22-26088Human embryonic stem cells secrete macrophage migration inhibitory factor: A novel findingPLOS ONE

Dear Dr. zhao,

Forgive us for the time that has pasted you submitted your manuscript - it has been difficult to get reviews, so I have decided to proceed with one review.  After careful consideration, we feel that it has merit but does not fully meet PLOS ONE’s publication criteria as it currently stands. Therefore, we invite you to submit a revised version of the manuscript that addresses the points raised during the review process.

Please directly address the points raised by this reviewer, send include a copy of your responses and changes/additions to the manuscript. 

We look forward to receiving your revised manuscript.

Kind regards,

Michael Klymkowsky, Ph.D.

Academic Editor

PLOS ONE

2. Please provide abstract in English.

“Funding: This work was supported by a grant from National Natural Science Foundation of China (grant numbers 81860238); Hainan Provincial Natural Science Foundation of China (grant number 821RC694); Major Research and Development Project of Hainan Province of China (grant number ZDYF2018233); Scientific research projects in Colleges and Universities of Hainan Province of China; Project of Hainan Province Clinical Medical Center;

Author Contributions

Zhenqiang Zhao and Jingquan Li conceived and designed the work, took part in all the works. Yanzhao Wei and Xiaohan Zheng were responsible for experiments,statistical analysis and writing of the manuscript.Ting Huang,Shengtong Sun,Xufang Wei and Qibing Liu were responsible for data collection and statistical analysis.”

“This work was supported by a grant from National Natural Science Foundation of China (grant numbers 81860238); Hainan Provincial Natural Science Foundation of China (grant number 821RC694); Major Research and Development Project of Hainan Province of China (grant number ZDYF2018233); Scientific research projects in Colleges and Universities of Hainan Province of China; Project of Hainan Province Clinical Medical Center;”

“Funding: This work was supported by a grant from National Natural Science Foundation of China (grant numbers 81860238); Hainan Provincial Natural Science Foundation of China (grant number 821RC694); Major Research and Development Project of Hainan Province of China (grant number ZDYF2018233); Scientific research projects in Colleges and Universities of Hainan Province of China; Project of Hainan Province Clinical Medical Center;

Author Contributions

Zhenqiang Zhao and Jingquan Li conceived and designed the work, took part in all the works. Yanzhao Wei and Xiaohan Zheng were responsible for experiments,statistical analysis and writing of the manuscript.Ting Huang,Shengtong Sun,Xufang Wei and Qibing Liu were responsible for data collection and statistical analysis.”

6. Thank you for stating the following in your Competing Interests section: 

“NO authors have competing interests”

7. We note that you have stated that you will provide repository information for your data at acceptance. Should your manuscript be accepted for publication, we will hold it until you provide the relevant accession numbers or DOIs necessary to access your data. If you wish to make changes to your Data Availability statement, please describe these changes in your cover letter and we will update your Data Availability statement to reflect the information you provide.

8. PLOS ONE now requires that authors provide the original uncropped and unadjusted images underlying all blot or gel results reported in a submission’s figures or Supporting Information files. This policy and the journal’s other requirements for blot/gel reporting and figure preparation are described in detail at https://journals.plos.org/plosone/s/figures#loc-blot-and-gel-reporting-requirements and https://journals.plos.org/plosone/s/figures#loc-preparing-figures-from-image-files. When you submit your revised manuscript, please ensure that your figures adhere fully to these guidelines and provide the original underlying images for all blot or gel data reported in your submission. See the following link for instructions on providing the original image data: https://journals.plos.org/plosone/s/figures#loc-original-images-for-blots-and-gels.

9. PLOS requires an ORCID iD for the corresponding author in Editorial Manager on papers submitted after December 6th, 2016. Please ensure that you have an ORCID iD and that it is validated in Editorial Manager. To do this, go to ‘Update my Information’ (in the upper left-hand corner of the main menu), and click on the Fetch/Validate link next to the ORCID field. This will take you to the ORCID site and allow you to create a new iD or authenticate a pre-existing iD in Editorial Manager. Please see the following video for instructions on linking an ORCID iD to your Editorial Manager account: https://www.youtube.com/watch?v=_xcclfuvtxQ.

10. Please include captions for your Supporting Information files at the end of your manuscript, and update any in-text citations to match accordingly. Please see our Supporting Information guidelines for more information: http://journals.plos.org/plosone/s/supporting-information.

Reviewers' comments:

Reviewer's Responses to Questions

**Comments to the Author**

1. Is the manuscript technically sound, and do the data support the conclusions?

Reviewer #1: Partly

2. Has the statistical analysis been performed appropriately and rigorously? 

Reviewer #1: Yes

3. Have the authors made all data underlying the findings in their manuscript fully available?

Reviewer #1: Yes

4. Is the manuscript presented in an intelligible fashion and written in standard English?

Reviewer #1: No

5. Review Comments to the Author

Reviewer #1: This is an interesting and original paper that shows that MIF is also produced grom human embrionic stem cells.

The Authors also demonstrate that MIF acts in HESC via binding to CXCR2 and CXCR7 and not through convebtional receptors such as CD74 and CD44.

While the finding is original and of interest its biological meaning remains elusive.

I have several points of concerns on the paper as it stands:

1. have the Authors evaluated whether HESC also secrete MIF-2 (DDT) ? There is no mention to MIF2 (DDT) in the paper and this point is of particular interest (PMID 30439447; PMID 30682543). In anycase the existence and possible secretion of DDT from HESC needs to be discussed

2. For search of completeness the Authors should discuss that MIF is implicated in several autoimmune diseases (PMID 18721909)

3. The statement that MIF suppresses tumour formation needs to be restated as the action of MIF in oncogenesis is complex and often dicothomic and may also favour tumor formation and progression (PMID 32769903; PMID 32155795; PMID 31867451; PMID 3334425)

4. As several specific MIF inhibitors are being developed and a semispecific inhibitor, ibidulast is already in the clinical setting, the present finding of the Authors of MIf production from HESC and the discussed potentil role of MIF in embryogenesis makes it important to mention that MIF inhibition during pregnancy should be carefullly evaluated

5. the paper is too long and could be reduced by 20%

6. PLOS authors have the option to publish the peer review history of their article (what does this mean?). If published, this will include your full peer review and any attached files.

Reviewer #1: No

---

## [Author Response · Author response to Decision Letter 0]

4 May 2023

Response to Academic Editors

1.This revision has been revised according to the PLOS ONE style template.

2.English abstracts have been provided.

3.The funding information has been changed in Edit Manager and the grant number has been verified.

4.Author contribution modified to “Zhenqiang Zhao and Tan Wang conceived and designed the work, took part in all the works. Yanzhao Wei and Xiaohan Zheng were responsible for experiments, statistical analysis and writing of the manuscript. Ting Huang, Yuanji Zhong, Shengtong Sun, Xufang Wei and Qibing Liu were responsible for data collection and statistical analysis. The funders had no role in study design, data collection and analysis, decision to publish, or preparation of the manuscript.” Modified text is marked in blue.

5.The grant-related text in the manuscript has been removed. Our funding statement is amended as follows：

This work was supported by a grant from National Natural Science Foundation of China (grant number 81860238); Hainan Provincial Natural Science Foundation of China (grant number 821RC694); Major Research and Development Project of Hainan Province of China (grant number ZDYF2018233); Scientific research projects in Colleges and Universities of Hainan Province of China (grant number HYYS2021A19); Project of Hainan Province Clinical Medical Center. The funders had no role in study design, data collection and analysis, decision to publish, or preparation of the manuscript. Modified text is marked in blue.

6.“The authors have declared that no competing interests exist.” has stated in the cover letter. Modified text is marked in blue.

7.Data availability is modified to “All relevant data are within the paper and its Supporting information files.” Modified text is marked in blue.

8.blot/gel image data has been added to the supporting information

9.ORCID iD of the corresponding author has been added in the Editorial Manager.

10.The title of the supporting information has been added at the end of the manuscript. 

Responses to reviewers

1.This revision adds a discussion of DDT to the discussion section. Modified text is marked in blue.

2.This revision adds the relationship of MIF to autoimmune diseases to the discussion section. Modified text is marked in blue.

3.The role of MIF in tumorigenesis has been removed due to the deletion of the text mentioned in point 5.

4.Due to the role of MIF in embryonic stem cells, MIF suppression during pregnancy should be evaluated as well. Modified text is marked in blue.

5.The text of this paper has been appropriately abridged to make it more concise.

---

## [Decision Letter · Decision Letter 1]

26 Jun 2023

Human embryonic stem cells secrete macrophage migration inhibitory factor: A novel finding

PONE-D-22-26088R1

Dear Dr. zhao,

We’re pleased to inform you that your manuscript has been judged scientifically suitable for publication and will be formally accepted for publication once it meets all outstanding technical requirements.

Kind regards,

Michael Klymkowsky, Ph.D.

Academic Editor

PLOS ONE

Additional Editor Comments (optional):

Reviewers' comments:

Reviewer's Responses to Questions

**Comments to the Author**

1. If the authors have adequately addressed your comments raised in a previous round of review and you feel that this manuscript is now acceptable for publication, you may indicate that here to bypass the “Comments to the Author” section, enter your conflict of interest statement in the “Confidential to Editor” section, and submit your "Accept" recommendation.

Reviewer #1: (No Response)

2. Is the manuscript technically sound, and do the data support the conclusions?

Reviewer #1: Yes

3. Has the statistical analysis been performed appropriately and rigorously? 

Reviewer #1: Yes

4. Have the authors made all data underlying the findings in their manuscript fully available?

Reviewer #1: (No Response)

5. Is the manuscript presented in an intelligible fashion and written in standard English?

Reviewer #1: Yes

6. Review Comments to the Author

Reviewer #1: (No Response)

7. PLOS authors have the option to publish the peer review history of their article (what does this mean?). If published, this will include your full peer review and any attached files.

Reviewer #1: No

---

## [Editor Report · Acceptance letter]

11 Aug 2023

PONE-D-22-26088R1 

Human embryonic stem cells secrete macrophage migration inhibitory factor: A novel finding 

Dear Dr. Zhao:

I'm pleased to inform you that your manuscript has been deemed suitable for publication in PLOS ONE. Congratulations! Your manuscript is now with our production department. 

Kind regards, 

on behalf of

Dr. Michael Klymkowsky 

Academic Editor

PLOS ONE